# Learning to Search with MCTSnet

## Abstract

Planning problems are among the most important and well-studied problems in artificial intelligence. They are most typically solved by tree search algorithms that simulate ahead into the future, evaluate future states, and back-up those evaluations to the root of a search tree. Among these algorithms, Monte-Carlo tree search (MCTS) is one of the most general, powerful and widely used. A typical implementation of MCTS uses cleverly designed rules, optimised to the particular characteristics of the domain. These rules control where the simulation traverses, what to evaluate in the states that are reached, and how to back-up those evaluations. In this paper we instead *learn* where, what and how to search. Our architecture, which we call an *MCTSnet*, incorporates simulation-based search *inside* a neural network, by expanding, evaluating and backing-up a vector embedding. The parameters of the network are trained end-to-end using gradient-based optimisation. When applied to small searches in the well-known planning problem Sokoban, the learned search algorithm significantly outperformed MCTS baselines.

## 1 Introduction

Many success stories in artificial intelligence are based on the application of powerful tree search algorithms to challenging planning problems (Samuel, 1959; Knuth & Moore, 1975; Jünger et al., 2009). It has been well documented that planning algorithms can be highly optimised by tailoring them to the domain (Schaeffer, 2000). For example, the performance can often be dramatically improved by modifying the rules that select the trajectory to traverse, the states to expand, the evaluation function by which performance is measured, and the backup rule by which those evaluations are propagated up the search tree. Our contribution is a new search algorithm in which all of these steps can be learned automatically and efficiently. Our work fits into a more general trend of learning differentiable versions of algorithms.

One particularly powerful and general method for planning is Monte-Carlo tree search (MCTS) (Coulom, 2006; Kocsis & Szepesvári, 2006), as used in the recent *AlphaGo* program (Silver et al., 2016). A typical MCTS algorithm consists of several phases. First, it simulates trajectories into the future, starting from a root state. Second, it evaluates the performance of leaf states - either using a random rollout, or using an evaluation function such as a 'value network'. Third, it backs-up these evaluations to update internal values along the trajectory, for example by averaging over evaluations.

We present a neural network architecture that includes the same processing stages as a typical MCTS, but inside the neural network itself, as a dynamic computational graph. The key idea is to represent the internal state of the search, at each node, by a memory vector. The computation of the network proceeds forwards from the root state, just like a simulation of MCTS, using a *simulation policy* based on the memory vector to select the trajectory to traverse. The leaf state is then processed by an *embedding network* to initialize the memory vector at the leaf. The network proceeds backwards up the trajectory, updating the memory at each visited state according to a *backup network* that propagates from child to parent. Finally, the root memory vector is used to compute an overall prediction of value or action.

The major benefit of our planning architecture, compared to more traditional planning algorithms, is that it can be exposed to gradient-based optimisation. This allows us to replace every component of MCTS with a richer, learnable equivalent — while maintaining the desirable structural properties of MCTS such as the use of a model, iterative local computations, and structured memory. We jointly train the parameters of the evaluation network, backup network and simulation policy so as to optimise the overall predictions of the MCTS network (MCTSnet). The majority of the network is fully differentiable, allowing for efficient training by gradient descent. Still, internal action sequences

directing the control flow of the network cannot be differentiated, and learning this internal policy presents a challenging credit assignment problem. To address this, we propose a novel, generally-applicable approximate scheme for credit assignment that leverages the anytime property of our computational graph, allowing us to also effectively learn this part of the search network from data.

In the Sokoban domain, a classic planning task (Botea et al., 2003), we justify our network design choices and show that our learned search algorithm is able to outperform various model-free and model-based baselines.

## 2    RELATED WORK

There has been significant previous work on learning evaluation functions, using supervised learning or reinforcement learning, that are subsequently combined with a search algorithm (Tesauro, 1994; Baxter et al., 1998; Silver et al., 2016). However, the learning process is typically decoupled from the search algorithm, and has no awareness of how the search algorithm will combine those evaluations into an overall decision.

Several previous search architectures have learned to tune the parameters of the evaluation function so as to achieve the most effective overall search results given a specified search algorithm. The learning-to-search framework (Chang et al., 2015) learns an evaluation function that is effective in the context of beam search. Samuel's checkers player (Samuel, 1959), the TD(leaf) algorithm (Baxter et al., 1998; Schaeffer et al., 2001), and the TreeStrap algorithm apply reinforcement learning to find an evaluation function that combines with minimax search to produce an accurate root evaluation (Veness et al., 2009); while comparison training (Tesauro, 1988) applies supervised learning to the same problem; these methods have been successful in chess, checkers and shogi. In all cases the evaluation function is scalar valued.

There have been a variety of previous efforts to frame the learning of internal search decisions as a meta-reasoning problem, one which can be optimized directly (Russell, 1995). Kocsis et al. (2005) apply black-box optimisation to learn the meta-parameters controlling an alpha-beta search, but do not learn fine-grained control over the search decisions. Considering action choices at tree nodes as a bandit problem led to the widely used UCT variant of MCTS (Kocsis & Szepesvári, 2006). Hay & Russell (2011) also studied the meta-problem in MCTS, but they only considered a myopic policy without function approximation. Pascanu et al. (2017) also investigate learning-to-plan using neural networks, but their approach is not differentiable, potentially limiting its scalability.

Other neural network architectures have also incorporated Monte-Carlo simulations. The I2A architecture (Weber et al., 2017) aggregates the results of several simulations into its network computation. MCTSnets both generalise and extend some ideas behind I2A: introducing a tree structured memory that stores node-specific statistics; and learning the simulation and tree expansion strategy, rather than rolling out each possible action from the root state with a fixed policy. Similar to I2A, the predictron architecture (Silver et al., 2017b) also aggregates over multiple simulations; however, in that case the simulations roll out an implicit transition model, rather than concrete steps from the actual environment.

## 3    MCTSNET

The MCTSnet architecture may be understood from two distinct but equivalent perspectives. First, it may be understood as a search algorithm with a control flow that closely mirrors the simulation-based tree traversals of MCTS. Second, it may be understood as a neural network represented by a computation graph that processes input states, performs intermediate computations on hidden states, and outputs a final decision. We present each of these perspectives in turn, starting with the original, unmodified search algorithm.

### 3.1    MCTS ALGORITHM

The goal of planning is to find the optimal strategy that maximises the total reward in an environment defined by a deterministic transition model $s' = T(s, a)$, mapping each state and action to a successor state $s'$, and a reward model $r(s, a)$, describing the goodness of each transition.

MCTS is a simulation-based search algorithm that converges to a solution to the planning problem. At a high level, the idea of MCTS is to maintain statistics at each node, such as the visit count and mean evaluation, and uses these statistics to decide which branches of the tree to visit.

MCTS proceeds by running a number of simulations. Each simulation traverses the tree, selecting the most promising child according to the statistics, until a leaf node is reached. The leaf node is then evaluated using a rollout or value-network (Silver et al., 2016). This value is then propagated during a back-up phase that updates statistics of the tree along the traversed path, tracking the visit counts $N(s)$, $N(s, a)$ and mean evaluation $Q(s, a)$ following from each state $s$ and action $a$. Search proceeds in an *anytime* fashion: the statistics gradually become more accurate, and simulations focus on increasingly promising regions of the tree.

We now describe a value-network MCTS in more detail. Each simulation from the root state $s_A$ is composed of four stages:

---

**Algorithm 1: Value-Network Monte-Carlo Tree Search**

1. Initialize simulation time $t = 0$ and current node $s_0 = s_A$.
2. Forward simulation from root state. Do until we reach a leaf node ($N(s_t) = 0$):
    (a) Sample action $a_t$ based on *simulation policy*,
    $a_t \sim \pi(a|s_t, \{N(s_t), N(s_t, a), Q(s_t, a)\}; \theta_s)$,
    (b) the reward $r_t = r(s_t, a_t)$ and next state $s_{t+1} = T(s_t, a_t)$ are computed
    (c) Increment $t$.
3. Evaluate leaf node $s_L$ found at depth $L$.
    (a) Obtain value estimate $V(s_L)$,
    (b) Set $N(s_L) = 1$.
4. Back-up phase from leaf node $s_L$, for each $t < L$
    (a) Set $(s, a) = (s_t, a_t)$.
    (b) Update $Q(s, a)$ towards the Monte-Carlo return:

$$Q(s, a) \leftarrow Q(s, a) + \frac{1}{N(s, a) + 1} \left( \sum_{t'=t}^{L-1} \gamma^{t'-t} r_{t'} + \gamma^{L-t} V(s_L) - Q(s, a) \right)$$

    (c) Update visit counts $N(s)$ and $N(s, a)$: $N(s) \leftarrow N(s) + 1$, $N(s, a) \leftarrow N(s, a) + 1$

---

When the search completes, it selects the action at the root with the most visit counts. The simulation policy $\pi$ is chosen to trade-off exploration and exploitation in the tree. In the UCT variant of MCTS (Kocsis & Szepesvári, 2006), $\pi$ is inspired by the UCB bandit algorithm (Auer, 2002).[1]

### 3.2 MCTSNET: SEARCH ALGORITHM

We now present MCTSnet as a generalisation of MCTS in which the statistics being tracked by the search algorithm, the backup update, and the expansion policy, are all learned from data.

MCTSnet proceeds by executing simulations that start from the root state $s_A$. When a simulation reaches a leaf node, that node is expanded and evaluated to produce a value and/or other statistics. The back-up phase then updates statistics in each node traversed during the simulation. Specifically, the parent node statistics are updated to new values that depend on the child values and also on their previous values. Finally, the selected action is chosen according to the statistics at the root of the search tree.

Different sub-networks are responsible for each of the components of the search described in the previous paragraph. Internally, these sub-networks manipulate the memory statistics $h$ at each node of the tree, which now have a vector representation, $h \in \mathbb{R}^n$. An embedding network $h \leftarrow \epsilon(s; \theta_e)$ evaluates the state $s$ and computes initial 'raw' statistics. A simulation policy $a \sim \pi(\cdot|h; \theta_s)$ is used to select actions during each simulation, based on statistics $h$. A backup network $h_{\text{parent}} \leftarrow \beta(h_{\text{parent}}, h_{\text{child}}; \theta_b)$ updates and propagates the statistics up the search tree. Finally, an overall decision (or evaluation) is made by a readout network $a \leftarrow \rho(h_{s_A}; \theta_r)$.

---

[1] In this case, the simulation policy is deterministic, $\pi(s) = \arg\max_a Q(s, a) + c\sqrt{\log(N(s))/N(s, a)}$.

An algorithmic description of the search network follows in Algorithm 2: [2]

---

**Algorithm 2: MCTSnet**

For $m = 1 \ldots M$, do simulation:

1. Initialize simulation time $t = 0$ and current node $s_0 = s_A$.

2. Forward simulation from root state. Do until we reach a leaf node ($N(s_t) = 0$):

    (a) Sample action $a_t$ based on *simulation policy*, $a_t \sim \pi(a|h_{s_t}; \theta_s)$,

    (b) the reward $r_t = r(s_t, a_t)$ and next state $s_{t+1} = T(s_t, a_t)$ are computed

    (c) Increment $t$.

3. Evaluate leaf node $s_L$ found at depth $L$.

    (a) Initialize node statistics using the embedding network: $h_{s_L} \leftarrow \epsilon(s_L; \theta_e)$

4. Back-up phase from leaf node $s_L$, for each $t < L$

    (a) Using the backup network $\beta$, update the node statistic as a function of its previous statistics and the statistic of its child:

$$h_{s_t} \leftarrow \beta(h_{s_t}, h_{s_{t+1}}, r_t, a_t; \theta_b)$$

After $M$ simulations, readout network outputs a (real) action distribution from the root memory, $\rho(h_{s_A}; \theta_r)$.

---

### 3.3 MCTSNET: NEURAL NETWORK ARCHITECTURE

We now present MCTSnet as a neural network architecture. The algorithm described above effectively defines a form of tree-structured memory: each node $s_k$ of the tree maintains its own corresponding statistics $h_k$. The statistics are initialized by the embedding network, but otherwise kept constant until the next time the node is visited. They are then updated using the backup network. For a fixed tree expansion, this allows us to see MCTSnet as a deep residual network with numerous skip connections, as well as a large number of inputs - all corresponding to different potential future states of the environment (these inputs actually are images of the actual input $s_A$ through applications of the transition operator T). We describe MCTSnet again following this different viewpoint in this section.

It is useful to introduce an index for the simulation count $m$, so that the tree memory after simulation $m$ is the set of $h_s^m$ for all tree nodes $s$. Conditioned on a tree path $p^{m+1} = s_0, a_0, s_1, a_1, \cdots, s_L$ for simulation $m + 1$, the MCTSnet memory gets updated as follows:

1. For $t = L$:

$$h_{s_L}^{m+1} \leftarrow \epsilon(s_L) \tag{1}$$

2. For $t < L$:

$$h_{s_t}^{m+1} \leftarrow \beta(h_{s_t}^m, h_{s_{t+1}}^{m+1}, r(s_t, a_t), a_t; \theta_b) \tag{2}$$

3. For all other $s$:

$$h_s^{m+1} \leftarrow h_s^m \qquad \text{(simulation } m + 1 \text{ is skipped)} \tag{3}$$

The tree path that gates this memory update is sampled as:

$$p^{m+1} \sim P(s_0 a_0 \cdots s_L | h^m) \propto \prod_{t=0}^{L-1} \pi(a_t | h_{s_t}^m; \theta_s) \mathbb{1}[s_{t+1} = T(s_t, a_t)], \tag{4}$$

where $L$ is a random stopping time for the tree path defined by $(N^m(s_{L-1}) > 0, N^m(s_L) = 0)$. An illustration of this update process is provided in Fig. 1. Note that the computation flow of the MCTS network is not defined by the final tree, but by the order in which nodes are visited (the tree expansion). Furthermore, as defined, MCTSnet is a feed-forward network with single input (the initial state) and single output (the action probabilities). However, thanks to the tree-structured memory, MCTSnet naturally allows for partial replanning, in a fashion similar to MCTS. Assume that from root-state $s_A$, the MCTS network chooses action $a$, and transitions (in the real environment) to new state $s_A'$. We can initialize the MCTS network for $s_A'$ as the subtree rooted in $s_A'$, and initialize node statistics of the subtree to their previously computed values.

---

[2] For clarity, we omitted the update of visits $N(s)$, used to determine leaf nodes, which proceeds as in Alg. 1.

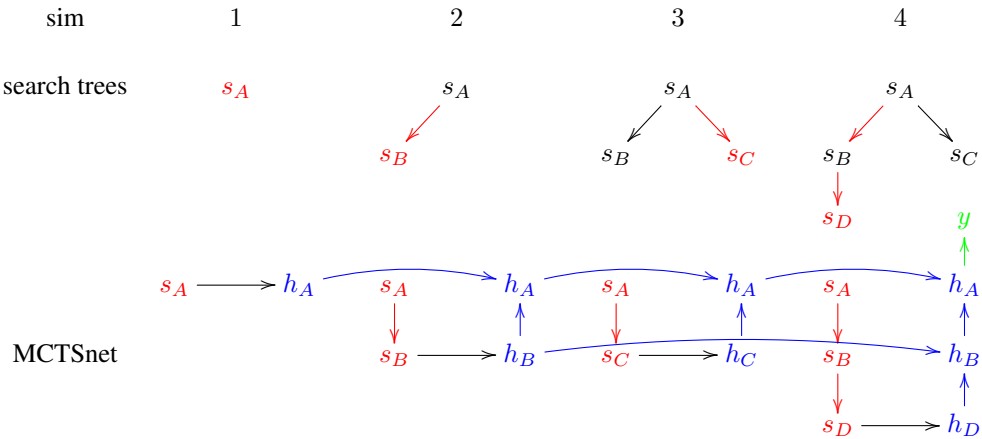

Figure 1: This diagram shows an execution of a search with $M = 4$. (Top) The evolution of the search tree rooted at $s_0$ after each simulation, with the last simulation path highlighted in red. (Bottom) The computation graph in MCTSnet resulting from these simulations. Black arrows represent the application of the **embedding network** $\epsilon(s)$ to initialize $h$ at tree node $s$. Red arrows represent the forward tree traversal during a simulation using the simulation policy (based on last memory state) and the environment model until a leaf node is reached. Blue arrows correspond to the backup network $\beta$, which updates the memory statistics $h$ along the traversed simulation path based on the child statistic and the last updated parent memory (in addition to transition information such as reward). The diagram makes it clear that this backup mechanism can skip over simulations where a particular node was not visited. For example, the fourth simulation updates $h_B$ based on $h_B$ from the second simulation, since $s_B$ was not visited during the third simulation. Finally, the readout network $\rho$, in green, outputs the action distribution based on the last root memory $h_A$. An expanded view of simulations is available in the appendix.

## 3.4 DESIGN CHOICES

We now provide design details for each sub-network in MCTSnet.

**Backup** $\beta$    The backup network contains a gated residual connection, allowing it to selectively ignore information originating from a node's subtree. It updates $h_S$ as follows:

$$\beta(\phi; \theta_b) = h_s + g(\phi; \theta_b)f(\phi; \theta_b), \tag{5}$$

where $\phi = (h_s, h_{s'}, r, a)$ and where $g$ is a learned gating function with range $[0, 1]$ and $f$ is the learned update function. We justify this architecture in Sec. 4.2, by comparing it to a simpler MLP which maps $\phi$ to the updated value of $h_s$.

**Learned simulation policy** $\pi$    In its basic, unstructured form, the simulation policy network is a simple MLP $k$ mapping the statistics $h_s$ to the logits $\psi(s, a)$, which define $\pi(a|s; \theta_s) \propto \exp(\psi(s, a))$.

We consider adding structure by modulating each logit with side-information corresponding to each action. One form of action-specific information is obtained from the child statistic $h_{T(s,a)}$. Another form of information comes from a learned policy prior $\mu$ over actions, with log-probabilities $\psi(s, a; \theta_p) = \log \mu(s, a; \theta_p)$; as in PUCT (Rosin, 2011). In our case, the policy prior comes from learning a small, model-free residual network on the same data. Combined, we obtain the following modulated network version for the simulation policy logits:

$$\psi(s, a) = w_0 \psi_p(s, a) + w_1 u\left(k(h_s), h_{T(s,a)}\right). \tag{6}$$

**embedding** $\epsilon$ **and readout network** $\rho$    The embedding network is a standard residual convolution network. The readout network, $\rho$, is a simple MLP that transforms a memory vector at the root into the required output format, in this case an action distribution. See appendix for details.

## 3.5 TRAINING MCTSNET

The readout network of MCTSnet ultimately outputs an overall decision or evaluation from the entire search. This final output may in principle be trained according to any loss function, such as by value-based or policy gradient reinforcement learning.

However, in order to focus on the novel aspects of our architecture, we choose to investigate the MCTSnet architecture in a supervised learning setup in which labels are first generated according to a standard, high-quality MCTS, and then an MCTSnet is trained to predict those labels, but using far fewer simulations.

Specifically, in our experiments, we first use MCTS, using a value network but no policy prior (Silver et al., 2017a) to generate good quality trajectories. For each real state $s$ encountered in a trajectory, we record the action $a^*$ the MCTS algorithm ended up taking, to create a dataset of state-action pairs $(s, a^*)$. Our objective is to then train MCTSnet to predict the action $a^*$ from state $s$. We note that, if this process was iterated, it would be similar to prior policy iteration schemes (Silver et al., 2017a; Anthony et al., 2017).

We denote $z_m$ the set of actions sampled stochastically during the $m^{\text{th}}$ simulation; $z_{\leq m}$ the set of all stochastic actions taken up to simulation $m$, and $\mathbf{z} = z_{\leq M}$ the set of all stochastic actions. The number of simulations $M$ is either chosen to be fixed, or taken from a stochastic distribution $p_M$.

After performing the desired number of simulations, the network output is the action probability vector $p_\theta(a|s, \mathbf{z})$. It is a random function of the state $s$ due to the stochastic actions $\mathbf{z}$. We choose to optimize $p_\theta(a|s, \mathbf{z})$ so that the prediction is on average correct, by minimizing the average cross entropy between the prediction and the correct label $a^*$. For a pair $(s, a^*)$, the loss is:

$$\ell(s, a^*) = \mathbb{E}_{\mathbf{z} \sim \pi(\mathbf{z}|\mathbf{s})} \left[ -\log p_\theta(a^*|s, \mathbf{z}) \right]. \tag{7}$$

This can also be interpreted as a lower-bound on the log-likelihood of the marginal distribution $p_\theta(a|s) = \mathbb{E}_{z \sim \pi(\mathbf{z}|\mathbf{s})} \left( p_\theta(a|\mathbf{z}, s) \right)$.

We minimize $l(s, a^*)$ by computing a single sample estimate of its gradient (Schulman et al., 2015):

$$\nabla_\theta \, \ell(s, a^*) = -\mathbb{E}_z \left[ \nabla_\theta \log p_\theta(a^*|s, \mathbf{z}) + (\nabla_\theta \log \pi(\mathbf{z}|s; \theta_s)) \log p_\theta(a^*|s, \mathbf{z}) \right]. \tag{8}$$

The first term of the gradient corresponds to the differentiable path of the network as described in section. The second term corresponds to the gradient with respect to the simulation distribution, and uses the REINFORCE or score-function method. In this term, the final log likelihood $\log p_\theta(a^*|s, \mathbf{z})$ plays the role of a 'reward' signal: in effect, the quality of the search is determined by the confidence in the correct label (as measured by its log-likelihood); the higher that confidence, the better the tree expansion, and the more the stochastic actions $\mathbf{z}$ that induced that tree expansion will be reinforced. In addition, we follow the common method of adding a neg-entropy regularization term on $\pi(a|s; \theta_s)$ to the loss, to prevent premature convergence.

### 3.6 A CREDIT ASSIGNMENT TECHNIQUE FOR ANYTIME ALGORITHMS

Although it is unbiased, the REINFORCE gradient above has very high variance; this is due to the difficulty of credit assignment in this problem: the number of decisions that contribute to a single decision $a^*$ is large (between $O(M \log M)$ and $O(M^2)$ for $M$ simulations), and understanding how each decision contributed led to a low error through a better tree expansion structure is very intricate.

In order to address this issue, we design a novel credit assignment technique for anytime algorithms, by casting the loss minimization for a single example as a sequential decision problem, and using reinforcement learning technique to come up with a family of estimators, allowing us to manipulate the bias-variance trade-off.

Consider a general anytime algorithm which, given an initial state $s$, can run for an arbitrary number of internal steps $M$ — in MCTSnet, these are simulations. For each step $m = 1 \dots M$, any number of stochastic decisions (collectively denoted $z_m$) may be taken, and at the end of each step, a candidate output distribution $p_\theta(a|s, z_{\leq m})$ may be evaluated against a loss function $\ell$. The value of the loss at the end of step $m$ is denoted $\ell_m \triangleq \ell(p_\theta(a|s, z_{\leq m}))$. We assume the objective is to maximize the terminal negative loss $-\ell_M$. Letting $\ell_0 = 0$, we rewrite the terminal loss as a telescoping sum:

$$-\ell_M = -(\ell_M - \ell_0) = \sum_{m=1\dots M} -(\ell_m - \ell_{m-1}) = \sum_{m=1\dots M} \bar{r}_m,$$

where we define the reward $\bar{r}_m$ as the decrease in loss $-(\ell_m - \ell_{m-1})$ obtained during the $m^{\text{th}}$ step. We then define the return $R_m = \sum_{m' \geq m} \bar{r}_{m'}$ as the sum of future rewards from step $m$; by definition we have $R_1 = -\ell_M$.

The REINFORCE term of equation (8) can be rewritten:

$$-\nabla_\theta \log \pi(\mathbf{z}|s;\theta_s) \log p_\theta(a^*|s,\mathbf{z}) = \sum_m \nabla_\theta \log \pi(z_m|s;\theta_s)R_1. \qquad (9)$$

Since stochastic variables in $z_m$ can only affect future rewards $r'_m, m' \geq m$, it follows from a classical policy gradient argument that (9) is, in expectation, also equal to:

$$\sum_m \nabla_\theta \log \pi(z_m|s, z_{<m};\theta_s)R_m = -\sum_m \nabla_\theta \log \pi(z_m|s, z_{<m};\theta_s)(\ell_M - \ell_{m-1}). \qquad (10)$$

In other words, the stochastic decisions from step $m$ do not simply use the terminal loss as reinforcement signal, but rather the difference between the terminal loss, and the loss computed before step $m$ started (the baseline). This estimate is still unbiased, but has lower variance, especially for the later steps of algorithm. Next, we trade off bias and variance by introducing a discount term $\gamma$. In essence, for simulation choices $z_m$, we choose to reward short term improvements more than later ones, since the relation between simulation $m$ and later improvements is harder to ascertain and likely to mostly appear as noise. Letting $R_m^\gamma = \sum_{m' \geq m} \gamma^{m'-m} r_{m'}$, our final gradient estimate of the MCTSnet loss becomes:

$$\nabla_\theta \, l(s, a^*) = \mathbb{E}_z \left[ -\nabla_\theta \log p_\theta(a^*|x, z) + \sum_m \nabla_\theta \log \pi(z_m|s;\theta_s)R_m^\gamma \right], \qquad (11)$$

$R_m^\gamma$ can be rewritten as the average of future baselined losses $l_{m+t} - l_{m-1}$, where $t$ follows a truncated geometric distribution with parameter $\gamma$ and maximum value $M - m$. Letting $\gamma = 0$ leads to a greedy behavior, where actions of simulation $m$ are only chosen as to maximize the immediate improvement in loss $-(\ell_m - \ell_{m-1})$. This myopic mode is linked to the *single-step assumption* proposed by Russell & Wefald (1989) in an analog context.

## 4 EXPERIMENTS

We investigate our architecture in the game of Sokoban, a classic, challenging puzzle game (Botea et al., 2003). As described above, our results are obtained in a supervised training regime. However, we continuously evaluate our network during training by running it as an agent in random Sokoban levels and report its success ratio in solving the levels. Throughout this experimental section, we keep the architecture and size of both embedding and readout network fixed, as detailed in the appendix.

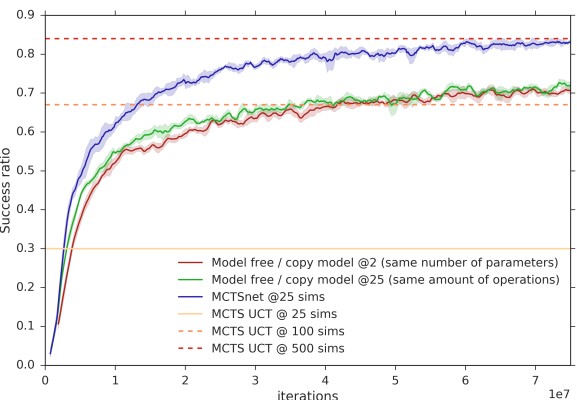

Figure 2: Evolution of success ratio in Sokoban during training using a continuous evaluator. MCTSnet (with $M = 25$) against two model-free copy-model baselines. In one case ($M = 2$), the copy-model has access to the same number of parameters and the same subnetworks. When $M = 25$, the baseline also matches the amount of computation. We also provide performance of MCTS with UCT with variable number of simulations.

### 4.1 MAIN RESULTS

We first compare our MCTSnet architecture with $M = 25$ simulations to a couple of model-free baselines. To assess whether the MCTSnet leverages the information contained in the simulations (from transition model $T$ and reward function $r$), we consider a version of the network that uses a sham environment model where $T(s, a) = s$ and $r(s, a) = 0$, but otherwise has identical architecture. For the case $M = 2$, the baseline has the same number of parameters as MCTSnet, and uses each subnetwork exactly once. We also test this architecture for the case $M = 25$, in which case the model-free baseline has the same number of parameters and can perform the same amount of computation but does not have access to the environment model — it is effectively model-free. We also evaluate a standard model-based

method (without learning) in the same environment: MCTS with a pre-learned value function (see appendix for details). When given access to 25 simulations per step, same as MCTSnet, we observe $\approx 30\%$ success ratio for MCTS in Sokoban. It requires 20 more times simulations for this version of MCTS to reach the level of performance of MCTSnet.

Overall, MCTSnet performs favorably against both model-based and model-free baselines, see Fig. 2. These comparisons validate two ingredients of our approach. First, the comparison of MCTSnet to its model-free variant confirms that it extracts information contained in states visited (and rewards obtained) during the search - in section 4.3 we show that it is also able to learn nontrivial search policies. Second, at test time, MCTSnet and MCTS both use the same environment model, and therefore have in principle access to the same information. The higher performance of MCTSnet demonstrates the benefits of learning and propagating vector-valued statistics which are richer and more informative than those tracked by MCTS.

Using the architecture detailed in Sec. 3.4 and 25 simulations, MCTSnets reach $84 \pm 1\%$ of levels solved[3] — close to the $87\%$ obtained in (Weber et al., 2017), although in a different setting (supervised vs reinforcement learning, 1e8 vs. 1e9 environment steps). We now consider more detailed comparisons to justify and understand our different design choices for MCTSnet.

## 4.2 LEARNED STATISTICS AND HOW TO PERFORM BACKUPS

In this section, we justify the backup network choice made in Sec. 3.4, by comparing the simple MLP version to the gated residual architecture we suggested. We find the gated residual architecture for the backup network to be advantageous both in terms of stability and accuracy. As Fig. 3 illustrates, with $M = 10$ simulations, the gated residual version systematically achieves better performance. For larger number of simulations ($M > 25$), we found the non-residual backup network to be simply numerically unstable.

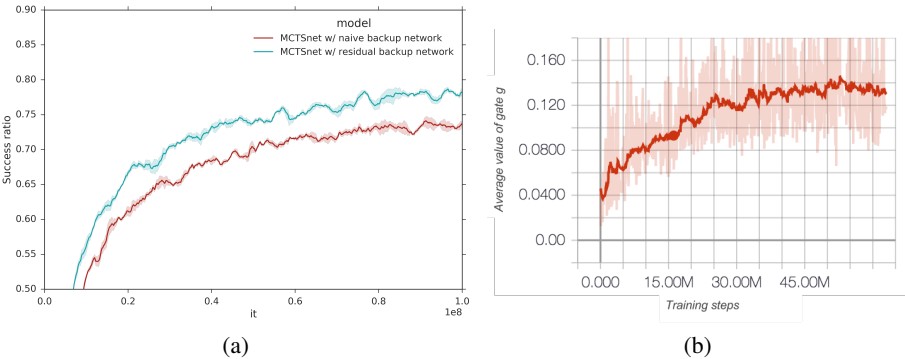

(a)                                             (b)

Figure 3: a) Comparison of backup network architectures. b) Typical evolution of the average gate value in the gated residual backup network. Initially, the gate prefers to minimize the influence of the memory update, it later gradually increases to take into account information from the subtree.

This can be explained by a large number of updates being recurrently applied, which can cause divergence when the network has arbitrary form. Instead, the gated network can quickly reduce the influence of the update coming from the subtree, and then slowly depart from the identity skip-connection; typical dynamics for the gate are displayed in Fig. 3-b, which showcases this behavior. For all other experiments, we therefore employed the gated residual backup network in MCTSnet.

## 4.3 LEARNING THE SIMULATION POLICY

As previously mentioned, learning the simulation policy is challenging because of the noisy estimation of the pseudo-return for the selected sequence $\mathbf{z}$. We investigate the effectiveness of our proposed designs for $\pi$ (see Sec. 3.4) and of our proposed approximate credit assignment scheme for learning its parameters (see Sec. 3.6).

---

[3]A video of MCTSnet solving Sokoban levels is available at this url: `https://goo.gl/2Bu8HD`. It shows a visualisation of the search tree's principal variation at each step, and the evolution of the (log) probability of the selected action as a function of the number of simulations.

**Basic setting** Despite the estimation issues, we verified that we can nevertheless lift $\pi$, in its simple form, above the performance of a MCTSnet using a uniform random simulation policy. Note that MCTSnet with a random simulation strategy already performs reasonably since it can still take advantage of the learned statistics, backups, and readout. See blue and red curves in Fig. 4a for a comparison.

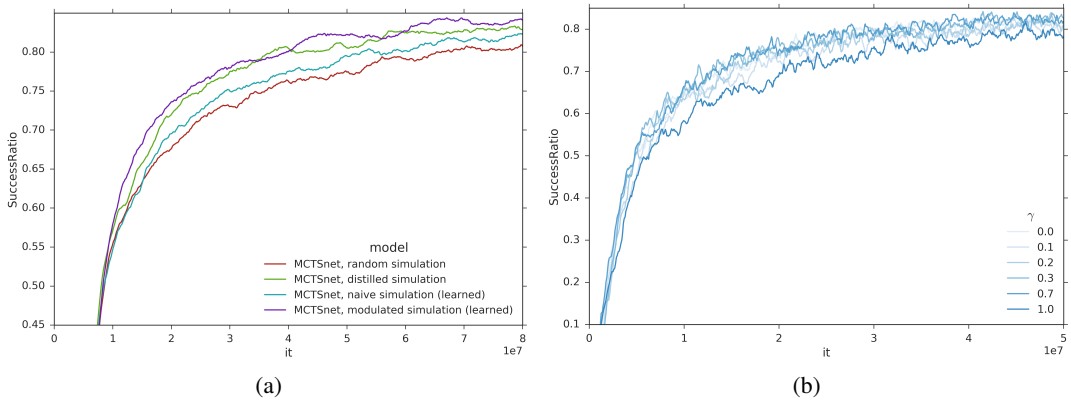

(a)                                                                      (b)

Figure 4: a) Comparison of different simulation policy strategies in MCTSnet with $M = 25$ simulations. b) Effect of different values of $\gamma$ in our approximate credit assignment scheme.

**Improved credit assignment technique** The vanilla gradient in Eq. (8) is enough to learn a simulation policy $\pi$ that performs better than a random search strategy, but it is still hampered by the noisy estimation process of the simulation policy gradient. A more effective search strategy can be learned using our proposed credit assignment scheme in Sec. 3.6 and the modulated policy architecture in Sec. 3.4. To show this, we train MCTSnets for different values of the discount $\gamma$ using the modulated policy architecture. The results in Fig. 4b demonstrate that the value of $\gamma = 1$, for which the network is optimizing the true loss, is not the ideal choice in MCTSnet. Lower values of $\gamma$ perform better at different stages of training. In late training, when the estimation problem is more stationary, the advantage of $\gamma < 1$ reduces but remains. The best performing MCTSnet architecture is shown in comparison to others in Fig. 4a. We also investigated whether simply providing the policy prior term in Eq. 6 (i.e., setting $w_1 = 0$) could match these results. With the policy prior learned with the right entropy regularization, this is indeed a well-performing simulation policy for MCTSnet with 25 simulations, but it did not match our best performing learned policy.

### 4.4 SCALABILITY WITH NUMBER OF SIMULATIONS

Thanks to weight sharing, MCTSnets can in principle be run for an arbitrary number of simulations $M \geq 1$. With larger number of simulations, the search has progressively more opportunities to query the environment model. To check whether our search network could take advantage of this during training, we compare MCTSnet for different number of simulations $M$, applied during both training and evaluation. In Fig. 5, we find that our approach was able to query and extract relevant information with additional simulations, generally achieving better results with less training steps.

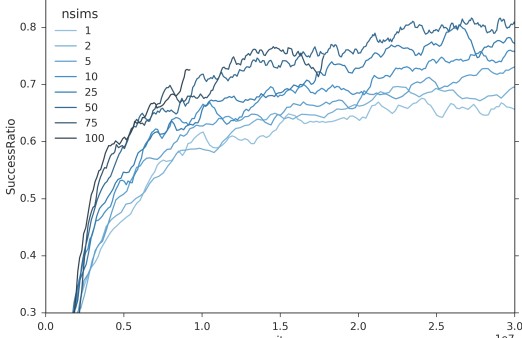

Figure 5: Performance of MCTSnets trained with different number of simulations $M$ (nsims). Generally, larger searches achieve better results with less training steps.

## 5 DISCUSSION

We have shown that it is possible to learn successful search algorithms by framing them as a dynamic computational graph that can be optimized

with gradient-based methods. This may be viewed as a first step towards a long-standing AI ambition: meta-reasoning about the internal processing of the agent. In particular, we proposed a neural version of the MCTS algorithm. The aim was to maintain the desirable properties of MCTS while allowing some flexibility to improve on the choice of nodes to expand, the statistics to store in memory, and the way in which they are propagated, all using gradient-based learning. A more pronounced departure from existing search algorithms could also be considered, although this may come at the expense of a harder optimization problem. We have also assumed that the true environment is available as a simulator, but this could also be relaxed: a model of the environment could be learned separately (Weber et al., 2017), or even end-to-end (Silver et al., 2017b). One advantage of this approach is that the search algorithm could learn how to make use of an imperfect model. Although we have focused on a supervised learning setup, our approach could easily be extended to a reinforcement learning setup by leveraging policy iteration with MCTS (Silver et al., 2017a; Anthony et al., 2017). We have focused on small searches, more similar in scale to the plans that are processed by the human brain (Arbib, 2003), than to the massive-scale searches in high-performance games or planning applications. In fact, our learned search performed better than a standard MCTS with more than an order-of-magnitude more computation, suggesting that neural approaches to search may ultimately replace their handcrafted counterparts.

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

## A ADDITIONAL FIGURES

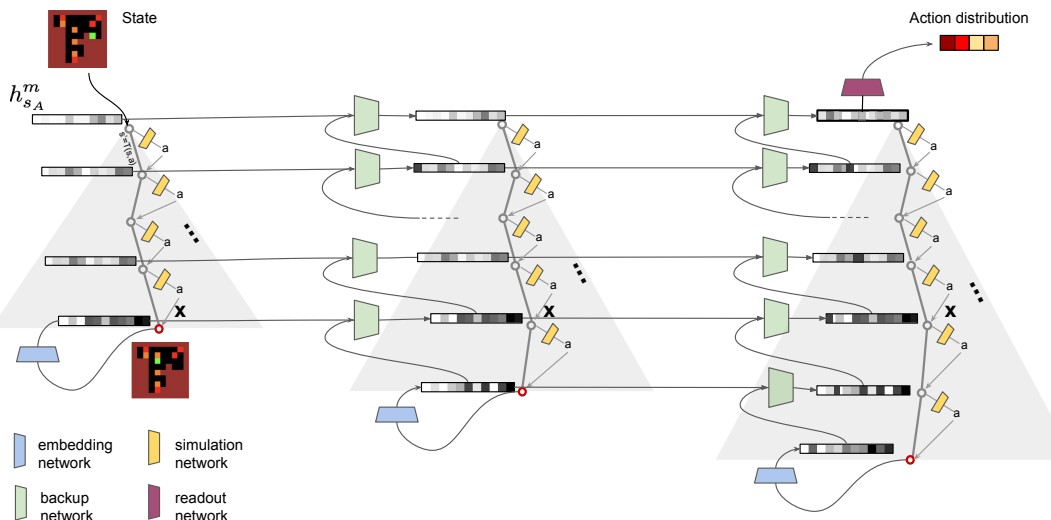

Figure 6: This diagram represents three simulations in MCTSnet that visit the tree node marked **x**. The leftmost simulation shows the first simulation $m$ during search to visit node **x** — when the memory statistic $h_{\mathbf{x}}^m$ is initialized by the embedding network. The middle simulation is the second simulation to traverse **x**, which may not immediately follow the leftmost simulation. The rightmost simulation is the third simulation to traverse **x**; it is also the final simulation overall, the readout network is employed to output the action distribution from the root memory statistic. This view showcases the skip-connection across simulation times to update the memory vectors. A diagram for a search with only two simulations is presented below.

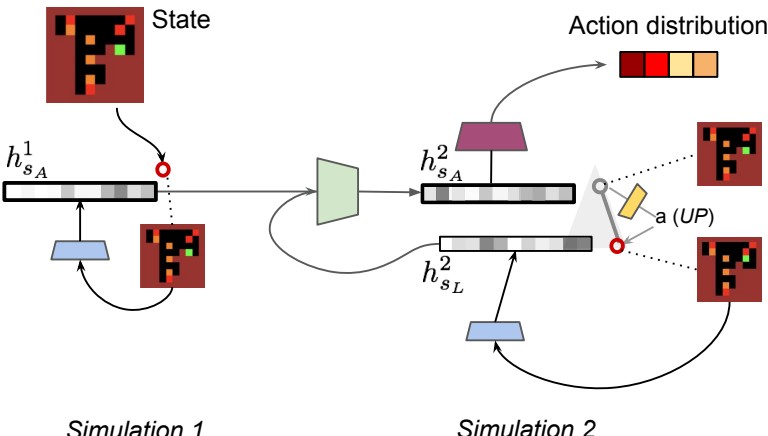

Figure 7: Diagram illustrating a MCTSnet search with exactly two simulations, using the same color codes for subnetworks as in Fig. 6.

## B ARCHITECTURAL CHOICES AND EXPERIMENTAL SETUP

For the Sokoban domain, we use $10 \times 10$ map layout with four boxes and targets. For level generation, we gained access to the level generator described by Weber et al. (2017). We directly provide a symbolic representation of the environment, coded as $10 \times 10 \times 4$ feature map (with one feature map per type of object: wall, agent, box, target), see Fig. 7 for a visual representation. The training dataset consists of 250000 trajectories of distinct levels; approximately 92% of those levels are solved by the agent; solved levels take on average 60 steps, while unsolved levels are interrupted after 100 steps. We also create a testing set with 2500 trajectories.

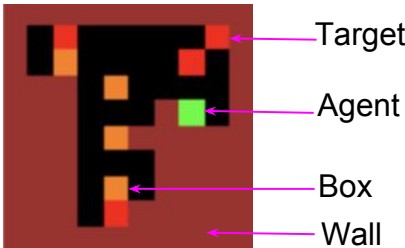

Figure 8: The different elements composing a Sokoban frame.

For MCTS (UCT variant), we use a pre-trained value network for leaf evaluation, depth-wise transposition tables to deal with symmetries, and we reuse the relevant search subtree after each real step.

### B.1 NETWORK ARCHITECTURE DETAILS

Our embedding network $\epsilon$ is a convolution network with 3 residual blocks. Each residual block is composed of two 64-channel convolution layers with 3x3 kernels applied with stride 1. The residual blocks are preceded by a convolution layer with the same properties, and followed by a convolution layer of 1x1 kernel to 32 channels. A linear layer maps the final convolutional activations into a 1D vector of size 128.

The readout network is a simple MLP with a single hidden layer of size 128. Non-linearities between all layers are ReLus. The policy prior network has a similar architecture to the embedding network, but with 2 residual blocks and 32-channel convolutions.

### B.2 TRAINING

We train MCTSnet using TensorFlow in an asynchronous distributed fashion. We use a batch of size 1, 32 workers and SGD for optimization with a learning rate of 5e-4.

