# OpenReview forum: "Learning to search with MCTSnets"
_ICLR.cc/2018/Conference — Reject_

### Official Review · AnonReviewer1 · 2017-11-28
**A solid, well-explained paper**

**Rating:** 7
**Confidence:** 3

**Review:**

The authors introduce an approach for adding learning to search capability to Monte Carlo tree search. The proposed method incorporates simulation-based search inside a neural network by expanding, evaluating and backing-up a vector-embedding. The key is to represent the internal state of the search by a memory vector at each node. The computation of the network proceeds just like a simulation of MCTS, but using a simulation policy based on the memory vector to initialize the memory vector at the leaf. The proposed method allows each component of MCTS to be rich and learnable, and allows the joint training of the evaluation network, backup network, and simulation policy in optimizing the MCTS network. The paper is thorough and well-explained. My only complaint is the evaluation is only done on one domain, Sokoban. More evaluations on diverse domains are called for.

---

> ### Author Response · Authors · 2017-12-19
> **response**
>
> Thank you for your review. We’ll look at more domains in future work.

---

### Official Review · AnonReviewer3 · 2017-11-28
**Some major concerns**

**Rating:** 4
**Confidence:** 4

**Review:**

This paper proposes a framework for learning to search, MCTSNet. The paper proposes an idea to integrate simulation-based planning into a neural network. By this integration, solving planning problems can be end-to-end training. The idea is to represent all operators such as backup, action selection and node initialisation by neural networks. The authors propose to train this using policy gradient in which data of optimal state-action pairs is generated by a standard MCTS with a large number of simulations.

In overall, it is a nice idea to use DNN to represent all update operators in MCTS. However the paper writing has many unclear points. In my point of view, the efficiency of the proposed framework is also questionable.


Here, I have many major concerns about the proposed idea.

- It looks like after training MCTSnet with a massive amount of data from another MCTS, MTCSnet algorithm as in Algorithm 2 will not do very much more planning yet. More simulations (M increases) will not make the statistics in 4(a) improved. Is it the reason why in experiments M is always small and increasing it does not make a huge improvement?. This is different from standard planning when a backup is handled with more simulations, the Q value function will have better statistics, and then get smaller regrets (see 4(b) in Algorithm 1). This supervising step is similar to one previous work [1] and not mentioned in the paper.


- MCTSnet has not dealt well with large/continuous state spaces. Each generated $s$ will amount to one tree node, with its own statistics. If M is large, the tree maintained is huge too. It might be not correct, then I am curious how this technical aspect is handled by MCTSnet.

- Other questions:

 + how the value network used in the MCTS in section 3.5 is constructed?

 + what does p(a|s,{\mathbf z}), p({\mathbf s}|{\mathbf z}) mean?

 + is R_1 similar to R^1

 + is z_m in section 3.5 and z^m in section 3.6 different?

 + is the action distribution from the root memory p_{\theta}(a|s)?

- Other related work:


[1] Xiaoxiao Guo et. al. Deep Learning for Real-Time Atari Game Play Using Offline Monte-Carlo Tree Search Planning, NIPS 2014

[2] Guez et. al. Bayes-Adaptive Simulation-based Search with Value Function Approximation, NIPS 2014

---

> ### Author Response · Authors · 2017-12-19
> **Response**
>
> Thank you for your review. We’re sorry you found some of the writing unclear. Let us answer point by point:
>
> * About the planning:
>
> The quality of the prediction does improve with more simulations up to M. And training with larger M does lead to better results, cf Figure 5.
>
> [1] distills search into a regular CNN. There is no learning to search. The fact that we generate labels for our experimental setup using search is orthogonal to our contribution. (Please also take a look at our answer to the other reviews.)
>
> The model-free baselines comparison provide evidence that well-chosen simulations of the environments are necessary to obtain good performance at test time. The data is used to learn how to perform such planning at test time.
>
> * About continuous spaces:
>
> This is irrelevant as we are not studying continuous environments. MCTS also suffers from the exact same limitation, but there exists techniques to deal with continuous states/actions that would also apply to MCTSnets. We could also “neuralize” search algorithms that are more directly suited to continuous spaces, if this had been the aim of the paper.
>
> *  “how the value network used in the MCTS in section 3.5 is constructed?”
>
> The value network is trained by regressing towards the outcome of rollouts of a pre-trained policy in Sokoban. We’ll add these details in the appendix.
>
> * “what does p(a|s,{\mathbf z}), p({\mathbf s}|{\mathbf z}) mean?”
>
>  p(a|s, z) is the output of the network, where z is a random variable representing the internal actions selected by the search.
>
>  p({\mathbf s}|{\mathbf z}) doesn’t appear in the paper. Do you mean \pi(z|s)? This is the distribution of z based on the simulation policy \pi for a given root state s; i.e. the probability of a tree expansion given the root state s.
>
> *  “is R_1 similar to R^1”
>
> Yes, R^1 should be R_1. This is a typo, thanks for catching that.
>
>  * “is z_m in section 3.5 and z^m in section 3.6 different?”
>
> That’s also a typo, they should be the same.
>
> *  “is the action distribution from the root memory p_{\theta}(a|s)?”
>
> That’s the marginal distribution after integrating over all random choices z.
>
> *  About other related work:
>
> None of these work attempt to learn how to search. We don’t view these as being particularly relevant.

---

### Official Review · AnonReviewer2 · 2017-11-28
**nice paper, flawed experiments**

**Rating:** 5
**Confidence:** 4

**Review:**

This paper designs a deep learning architecture that mimics the structure of the well-known MCTS algorithm. From gold standard state-action pairs, it learns each component of this architecture in order to predict similar actions.

I enjoyed reading this paper. The presentation is very clear, the design of the architecture is beautiful, and I was especially impressed with the related work discussion that went back to identify other game search and RL work that attempts to learn parts of the search algorithm. Nice job overall.

The main flaw of the paper is in its experiments. If I understand them correctly, the comparison is between a neural network that has been learned on 250,000 trajectories of 60 steps each where each step is decided by a ground truth close-to-optimal algorithm, say MCTS with 1000 rollouts (is this mentioned in the paper). That makes for a staggering 15 billion rollouts of prior data that goes into the MCTSNet model. This is compared to 25 rollouts of MCTS that make the decision for the baseline. I suspect that generating the training data and learning the model takes an enormous amount of CPU time, while 25 MCTS rollouts can probably be done in a second or two. I'm sure I'm misinterpreting some detail here, but how is this a fair comparison?

Would it be fair to have a baseline that learns the MCTS coefficient on the training data? Or one that uses the value function that was learned with classic search? I find it difficult to understand the details of the experimental setup, and maybe some of these experiments are reported. Please clarify. Also: colors are not distinguishable in grey print.

How would the technique scale with more MCTS iterations? I suspect that the O(N^2) complexity is very prohibitive and will not allow this to scale up?

I'm a bit worried about the idea of learning to trade off exploration and exploitation. In the end you'll just allow for the minimal amount of exploration to solve the games you've already seen. This seems risky, and I suspect that UCB and more statistically principled approaches would be more robust in this regard?

Are these Sokoban puzzles easy for classical AI techniques? I know that many of them can be solved by A* search with a decent heuristic. It would be fair to discuss this.

The last sentence of conclusions is too far reaching; there is really no evidence for that claim.

---

> ### Author Response · Authors · 2017-12-19
> **Response**
>
> Thank you for your thoughtful review. Let us reply to each point separately.
>
> * About the experimental setup:
> It is true that the network is trained from a ground-truth close-to-optimal MCTS that uses a lot of computation (1000 rollouts per search). But that is exactly the point! Our neural network can efficiently represent and learn a search strategy that would normally take 1000 rollouts of MCTS to compute. Whereas a standard neural network (of equivalent capacity) fails to learn the search strategy, even when given the same close-to-optimal MCTS training data. Also note our MCTS baseline uses a value network trained with a comparable amount of data. Finally - when solving a level never seen before, MCTS and MCTSnet have access to the exact same amount of model information.
>
> A valid suggestion you make is to “learn[..] the coefficients of MCTS”. MCTSnets is exactly such a proposal, albeit with a more general architecture.
>
> * About exploration/exploitation tradeoff:
>
> In the context of search, we care about exploration vs exploitation strategies inasmuch as they allow us to get good final action selection. UCB has proven empirically to be a good simulation strategy, but there is no guarantee it is optimal in that context.
>
> It is in principle possible for our networks to implicitly re-discover UCB if it is was indeed optimal for our problem. The advantage of learning is that we can tune that strategy to the domain at hand, instead of relying on a generic approach which might perform indifferently in the task of interest.
>
> We’ve shown some evidence that we learn an effective simulation strategy in our experimental setup; furthermore, we find that the trees constructed by MCTSnet contained variable number of branches; we believe this implies some form exploration / tradeoff is learned by the algorithm.
>
> * About scaling:
>
> Note that O(n^2) is a worse-case complexity (it will typically be O(n log n)) and not specific to MCTSnets, regular MCTS has the same complexity.
> MCTSnets compares favorably to MCTS (using deep nets) at run-time: they both run a forward of a large network for each simulation to evaluate leaf nodes. In addition, MCTSnet also runs a simulation policy and backup networks, but these are much less expensive in comparison.
>
> It’s training and optimizing MCTSnets with very large number of simulation that is more challenging. But the hope, partially demonstrated here, is that learned search can do more with less simulations - so very large number of sims might not be required.
>
> * About sokoban puzzles:
>
> We’re using Sokoban as a sandbox for our model. We are not attempting to compete with brute-force solvers using heuristics and other domain-specific knowledge on Sokoban - we are not putting forward our method as a Sokoban solver. Indeed, a well-tuned A* may do relatively well.
>
> * About last sentence in conclusion:
>
> We can rephrase. But in the paper we do demonstrate that MCTSnet, with its custom rules, outperforms MCTS, with its handcrafted one. In our mind, this justifies the suggestion that learning search rules may outperform hand-crafted ones. We don’t view this as too controversial that this is a possibility, and it has been suggested before (cf meta-control literature).

---

### Decision · Program_Chairs · 2018-01-29
**ICLR 2018 Conference Acceptance Decision**

**Decision:**

Reject

**Comment:**

All reviewers agree that the contribution of this paper, a new way of training neural nets to execute Monte-Carlo Tree Search, is an appealing idea.  For the most part, the reviewers found the exposition to be fairly clear, and the proposed architecture of good technical quality.  Two of the reviewers point out flaws in implementing in a single domain, 10x10 Sokoban with four boxes and four targets.  Since their training methodology uses supervised training on approximate ground-truth trajectories derived from extensive plain MCTS trials, it seems unlikely that the trained DNN will be able to generalize to other geometries (beyond 10x10x4) that were not seen during training.  Sokoban also has a low branching ratio, so that these experiments do not provide any insight into how the methodology will scale at much higher branching ratios.

Pros: Good technical quality, interesting novel idea, exposition is mostly clear.  Good empirical results in one very limited domain.
Cons: Single 10x10x4 Sokoban domain is too limited to derive any general conclusions.

Point for improvement: The paper compares performance of MCTSnet trials vs. plain MCTS trials based on the number of trials performed.  This is not an appropriate comparison, because the NN trials will be much more heavyweight in terms of CPU time, and there is usually a time limit to cut off MCTS trials and execute an action.  It will be much better to plot performance of MCTSnet and plain MCTS vs. CPU time used.